# Factors Affecting Participation in Cervical Screening by Female Nurses in Public Health Institutions in Vhembe District, Limpopo Province

Lindelani Mathivha, Dorah Ursula Ramathuba * and Maria Sonto Maputle

Department of Advanced Nursing Science, Faculty of Health Science at the University of Venda Limpopo Province and South Africa, University of Venda, Thohoyandou 0950, South Africa
* Correspondence: dorah.ramathuba@univen.ac.za

**Abstract:** In South Africa, the prevalence and death rates as a result of cervical cancer remains high, creating social and economic instability. The main objective of this study was to determine factors affecting participation in cervical screening by female nurses in public health institutions in Vhembe district, Limpopo Province. Early diagnosis and treatment are essential in cervical cancer screening as the prevalence of the disease decreases. The study was carried out at public health institutions in Vhembe district, Limpopo Province. A quantitative, descriptive, cross-sectional design was used in this study. Structured self-reported questionnaires were used in the collection of data. Descriptive statistics were used when analysing data to identify statistically significant differences in variables using SPSS version 26, and the findings were presented in percentages to generate evidence for the study. According to the study findings, (218, 83%) female nurses had been screened for cervical cancer, while the minority (46, 17%) had not been screened. The reasons cited were that they thought they were healthy (82, 31%), (79, 30%) felt embarrassed, and (15%) feared positive results. The majority (190) of them had last been screened more than three years before, with only a few (27, 10%) screened within the previous three years. A hundred and forty-two (53.8%) displayed negative attitudes and practices towards screening if it is paid for, and a hundred and eighteen (44.6%) perceived themselves as not vulnerable to acquiring cervical carcinoma. Furthermore,(128, 48.5%) strongly disagreed and 17(6.4%) were undecided about being screened by a male practitioner. The study concluded that negative attitudes, poor perception, and embarrassment are factors leading to low uptake by female nurses. Therefore, this study recommends that the Department of Health should build the capacity of nursing staff on matters of national significance to achieve sustainable goals and be a healthy nation. Nurses should be at the forefront of departmental programmes.

**Keywords:** cervical cancer; factors; female nurses; participation; public health institutions and screening

## 1. Introduction

Cervical cancer is an escalating health problem and a significant cause of mortality across all regions of the world. [1]. Although cervical cancer is preventable, it is still endemic, affecting many women in impoverished and developing countries [2]. In South Africa, cervical cancer incidences have remained the same since 2002, as reported in the 2017 national prevention policy. According to [3], human papilloma virus causes 99% of all cervical cancer worldwide, and the provision of HPV vaccination is offered for its prevention. However, other co-factors have been found including having concurrent sex partners, becoming sexually active early, having more than five pregnancies, current or previous STIs including HIV, and active tobacco use [4].

According to [1], prompt diagnosis and treatment of premalignant lesions reduces the possible burden of cancer becoming invasive. The authors [5] indicated that attendance of cervical screening is below par in sub-Saharan countries, with limited healthcare service facilities and poor infrastructure laboratories.

Furthermore, the authors in [6] also indicated that other factors leading to the low uptake of cervical cancer screening includes poor knowledge about cervical cancer, poor attitudes towards disease, and lack of awareness of risk factors can affect screening practices and the improvement of preventive behaviours for cervical cancer.

According to the report in [7], health workers, especially nurses, also experience incidence of cervical cancer and deaths due to cervical cancer. In a study conducted in Qatar by [8], despite knowledge of cervical cancer and prevention among female nurses, their attitude towards cervical screening was negative. Furthermore, the study revealed that none of the female nurses who participated in the study had undergone cervical screening and the main reason was due to fear of embarrassment, fear of pain, and lack of interest [8,9]. Among other barriers to effective cervical cancer screening which were observed include a lack of knowledge, incompetent staff, not feeling at risk, lack of symptoms, negligence, being insensitive to examination, a lack of interest, and the test being unpleasant. This attitude and negative beliefs towards cervical cancer were the reason for poor screening behaviour.

In another study conducted in Ethiopia and Nigeria by the authors in [10,11], among 225 nurses who participated in the study, a minority were previously tested for cervical cancer. The most common reason given for not being screened included a lack of judgement (17.9%), worrying about positive results (16.4%), and feeling of discomfort (10.9%). As evidenced by the different literature, an improved level of knowledge and awareness demonstrated by nurses prompted them to participate in screening services. These factors were identified to be barriers and contributed to the low uptake of screening services.

Theories such as the Health Belief Model, which has been evolving since 1950s, were designed to assist and envisage why individuals monitor their health to prevent and control health conditions. This model supposes that health behaviours are encouraged by self-susceptibility, seriousness, benefits and barriers to behaviours, cues to action, and self–efficacy related to the disease. In the previous study by the author of [2], it was revealed that these elements are what motivates an individual to take health action if they understand the benefits of prevention.

It was further revealed that using the Health Belief Model established perceived benefits and barriers as an influence on early detection and it was proven that the Health Belief Model could affect decision making in the individual to screen for cervical cancer. Research concerning female nurses' participation in cervical cancer screening in South Africa is lacking. Research studies show the results of women in communities, but not the population of health care workers. Therefore, little is known about factors affecting participation in cervical screening by female nurses in public health institutions in Vhembe district, Limpopo Province, as there is a gap or lack of literature focusing on this phenomenon. The study may contribute insights to health care services, curriculum developers, and policy makers in evaluating the impact of screening services and promotion of the acceptance by health care workers to increase the uptake of cervical cancer screening services.

*Problem Statement*

Worldwide, cervical cancer is the fourth most common cancer affecting women and around 85% of the global burden occurs in the less-developed countries, where cervical cancer accounts for almost 12% of female cancers [12]. In South Africa, about 12,983 new cervical cancer cases were recorded in 2018, as reported by the National Cancer Registry (NCR), while there were 5785 new cases of cervical cancer in 2012 [13]. This indicates that the number showed a global increase. The author in [14] reported that there has been a decline in cervical screenings conducted annually in Limpopo Province from 82,041 in 2013 to 23,527 in 2015. Furthermore, the distribution of cervical cancer by geographical area in Limpopo Province shows statistically significant increases in cervical cancer prevalence and was observed in all five districts. Mopani district had the highest prevalence in 2013 and 2014, followed by Waterberg district in 2015 at 20%, 19% and 21.9%. Vhembe district

had the lowest cervical cancer prevalence compared to other four districts, but this district is showing an increasing trend, from 14.3% in 2013 to 15.9% in 2015.

The decline in cervical screening in Vhembe among women could results in high morbidity and mortality from this disease with a devastating impact on society. Cancer is responsible for the premature removal of many economically active women, mothers, and grandmothers from society. This poses not only financial burden on a family, but also social and emotional trauma to other family members, an alteration in family structure because young children must drop out of school to become caregivers, a loss of amenities, and a fall below poverty line.

## 2. Material and Methods

The study adopted a descriptive cross-sectional design to describe factors affecting participation in cervical screening by female nurses in public health institutions in Vhembe district, Limpopo Province. This approach assisted the researcher in gathering respondents' knowledge on cervical screening, attitudes, and practices related to cervical screening, as it was gathered on a wider or broader scale with the aid of structured questionnaires or formal instruments when the data were collected.

### 2.1. Study Setting

The study was carried out at four public health institutions which were selected for research purposes, including two hospitals (Tshilidzini Hospital and Donald Fraser Hospital) and two community health centres (Thohoyandou CHC and William Eddie CHC), which are found within Thulamela local area, in Vhembe region.

### 2.2. Population

The target population in this study were female nurses who were working in the public health institutions around the area in the Vhembe region of Limpopo province. The population was 595 nurses from all the four selected public health institutions.

### 2.3. Sampling

Purposive sampling was used to sample hospitals found in Vhembe district. Donald Fraser Hospital serves as a district hospital and Tshilidzini Hospital serves as regional hospital in the heart of the Vhembe district. Primary health clinics and community health centres around the Thulamela local area transfer their patients to these two hospitals. Other health care centres and facilities included in this study include Thohoyandou CHC and William Eddie CHC as they are found within this area, and these two facilities provide support to the district hospital and regional hospital. Inclusion criteria were nurses who were employed for more than five years, and in the age range of 25–65 years. Newly employed nurses were excluded.

The sample size was calculated using Slovin's formula and the total sample was 264 including 10% of non-response rate.

Then, finally, a simple random technique was employed to select female nurses to be part of the study from each facility. Each group of nurses from the four selected facilities was represented to ensure that there was no sampling bias. These groups of female nurses included registered nurses (RN), enrolled nurses (EN), and enrolled nursing auxiliaries (ENA).

### 2.4. Data Collection

Data collection was carried out after approval from the University Research Ethics Committee (SHS/21/PDC/07/0505) and authorization to undertake the study was approved by the Department of Health and the public health institutions involved in the study. Then, the researcher first met with managers within the hospital or health centre selected for permission to meet with respondents in their respective wards to explain the project to the respondents and what was expected from them. The idea of meeting before collecting data

was to provide information and ensure that there was full understanding for respondents to make independent decisions to participate without being coerced. The objective, aim, and benefits were outlined to the participants to get informed consent, and those willing to participate provided written consent to be part of the study. An appointment was made to visit the respondents during their lunch time. Respondents in this study were female nurses who worked at the public hospitals and in public health care centre.

*2.5. Measurement Instrument in Data Collection*

Data was collected with the aid of a structured questionnaire prepared in the English language. The reliability of the research instrument was evaluated through the use of a pre-test with fifteen nurses from an institution not included in the study to ascertain stability of the instrument and clarity of questions. The Cronbach alpha reliability coefficient was 0.70. The data collection instrument was developed considering the objective of the study.

The items consisted of socio-demographic characteristics (Section A), knowledge of cervical cancer screening (Section B), and the last item comprised attitudes and practices towards participation in cervical screening (Section C). In section B, the knowledge of female nurses was assessed and if respondent scores were <60%, they were considered as not knowledgeable about cervical cancer screening. Those respondents who scored ≥60% were considered knowledgeable. The attitude towards and practices of cervical cancer screening was assessed and respondents who scored <60% were considered to have a negative attitude, while those who score ≥60% were considered to have a positive attitude.

## 3. Data Analysis

Collected data was cleaned or verified before being entered into the system by the researcher. Questions with errors or that were incomplete were not entered into the system. A data technician was employed to capture verified data into a standardized format. The statistical analysis technique was used, namely the Social Package for Social Science (SPSS version 26). The results were presented in the form of tables and figures and interpreted descriptively so that it has meaning to the readers of the research report. Statistical comparison using chi-square was undertaken to see relationships between the variables including the knowledge of cervical cancer, and attitude and practice towards cervical screening, in female nurses.

## 4. Presentation of Results

Table 1 below displays results on demographic variables of the study respondents. According to the results on Table 1, most respondents (117, 44.3%) were drawn from the Donald Fraser Hospital, followed by 31.8% from the Tshilidzini Hospital, and the minority from Thohoyandou Health Centre (14.3%) and the William Eddie Hospital (9.8%), respectively. The age range of the study respondents was dominated by the 36–45 age group which constituted 39.8%, while 23.1% were below 25 years of age. However, the minority were between 26 and 35 years of age and those above 56 years and above age group, were (46, 17%). According to the data collected, it was revealed that the majority of the respondents were holders of a diploma (85, 32.2%), while the number of post-graduates was (47, 17.8%) and those with a certificate in nursing was also key. Regarding professional levels, the majority (177, 67%) of the respondents were registered professional nurses, followed by enrolled assistant nurses (50, 18.9%), and, lastly, the enrolled nurses who constituted 14%. The analysis of the participant's marital status revealed that the majority (213, 80.7%) were married, while the minority (36, 13.6%) were divorced followed by (15, 5.7%) widowed individuals. In terms of parental status, many of the study respondents had five children and above, with (77, 29.2%) having one to two children, and (67, 25.4%) having three to four children; however, (37, 14%) had never had children in their life. Although the respondents were Africans, a high proportion of participants (196, 74.2%) were Christians, whereas (68, 25.8%) of the participants were from the African Orthodox faith.

**Table 1.** Demographic information of respondents.

| Characteristics | Frequency (n) | Percentage (%) | Frequency (n) | Percentage (%) | Frequency (n) | Percentage (%) |
|---|---|---|---|---|---|---|
| **Health Facility** | | | **Age of respondents** | | **Professional Qualifications** | |
| Thohoyandou Health Centre | 37 | 14.03% | 25–30 years | 61 (23.1%) | Diploma | 85 (32.2%) |
| Tshilidzini Hospital | 83 | 31.8% | 31–35 years | 17(6.4%) | Degree | 80 (30.3%) |
| Donald Fraser Hospital | 117 | 44.3% | 36–45 years | 105 (39.8%) | Postgraduate | 47 (17.8%) |
| William Eddie Hospital | 26 | 9.8% | 46–55 years | 36(13.6%) | Certificate | 52(19.7%) |
| Total | 264 | 100.0% | 56 years and above | 45 (17.0%) | Total | 264 (100.0%) |
| | | | Total | 264 (100.0) | | |
| **Professional level** | | | **Duration of Service** | | **Marital Status** | |
| Registered Professional Nurse | 177 | 67.0% | Less than 1 Year | 8 (3.0%) | Married | 213 (89.7%) |
| Enrolled Nurse | 37 | 14.0% | 1 to 3 years | 34 (12.9%) | Divorce | 36 (13.6%) |
| Enrolled Nursing Assistant | 50 | 18.9% | 3 to 5 years | 14 (5.3%) | Widowed | 15 (5.7%) |
| Total | 264 | 100.0% | 5 years and above | 208 (78.8%) | Total | 264 (100.0%) |
| | | | Total | 264 (100.0%) | | |
| **Parental Status** | | | **Religion** | | | |
| Nullipara | 37 | 14.0% | Christianity | | 196 | 74.2% |
| 1 to 2 children | 77 | 29.2% | Orthodox | | 68 | 25.8% |
| 3 to 4 children | 67 | 25.4% | Total | | 264 | 100.0% |
| 5 children and above | 83 | 31.4% | | | | |
| Total | | 100.0% | | | | |

### 4.1. Sources of Information Regarding Cervical Cancer and Screening

Findings revealed that majority of respondents obtained information through electronic media (61, 23.1%), training (58, 22%), and from the doctor (57, 21.6%). However, the minority established that they heard it through their friend (39, 14.8%), hospital talks (7, 10.2%), and from print media (22, 8.3%). Sources of information are varied now with more people having access to technology and electronic media.

### 4.2. Risk Factors Regarding Cervical Cancer

As shown in Figure 1 below, the findings established that the high percentage identified early sexual intercourse (89, 34%), acquiring the HPV virus (64, 24%), and having numerous sexual partners (61, 23%) as the major risk factors of cervical cancer. However, (40, 15%) female nurses indicated that they did not know the cause and the least identified cigarette smoking. Risks factors of cervical cancer are usually associated with sex-related diseases rather than other personal factors.

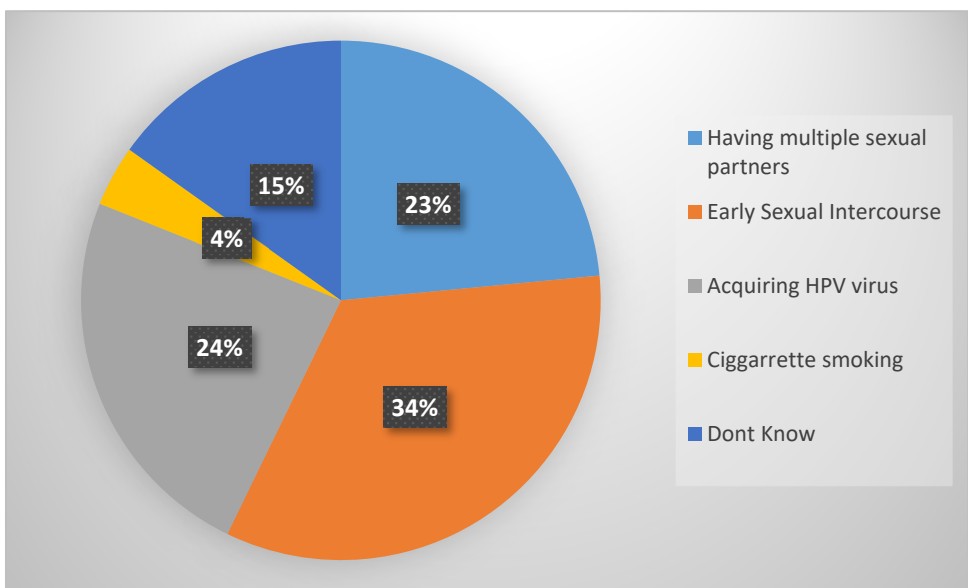

**Figure 1.** Risk factors of cervical cancer.

### 4.3. Knowledge Regarding Signs and Symptoms of Cervical Cancer

Findings revealed that majority (158, 60%) of respondents indicated vaginal bleeding as the main symptom followed by post-menopausal bleeding (32, 12%), contact bleeding (24, 9%), and a lastly foul vaginal smell (14, 5.3%). Female nurses had more knowledge with regard to the signs and symptoms of cervical cancer.

### 4.4. Knowledge Regarding Prevention of Cervical Cancer

As shown in Figure 2 below, the findings show that most (143, 54.2%) respondents were not informed about the prevention methods of cervical cancer. However, only a few identified HPV vaccination (53, 20%), the avoidance of multiple sexual partners (35, 13.3%), the avoidance of early sexual intercourse (20, 7.6%), as well as the quitting of cigarette smoking (13, 4.9%) as methods to prevent cervical cancer. Knowledge is power and more awareness is more important. This can be achieved by information dissemination by health workers and community health care workers.

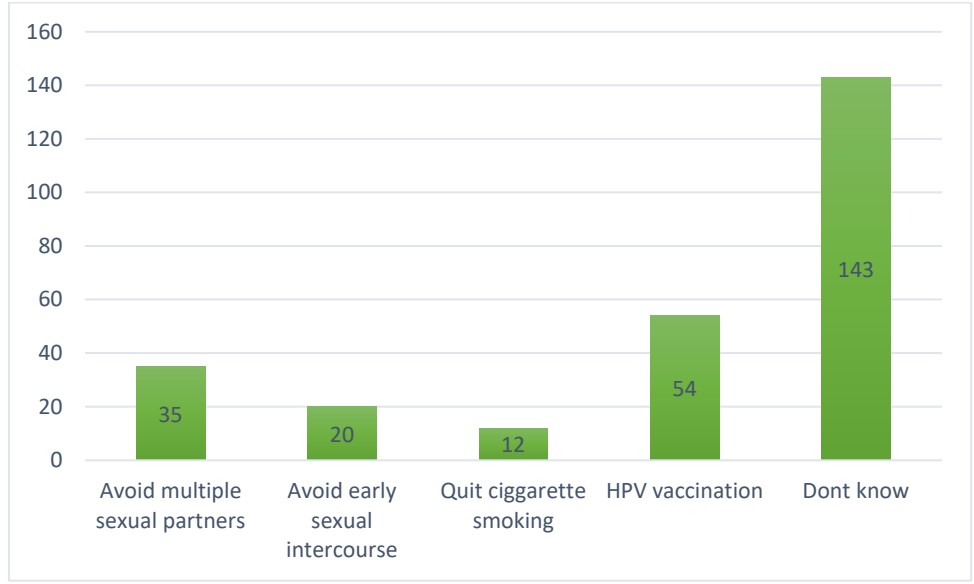

**Figure 2.** Prevention of cervical cancer.

### 4.5. Knowledge Regarding Methods of Cervical Cancer Screening and Frequency

The well-known methods for screening cervical cancer were revealed by many (209, 79%) as the pap smear; however, (53, 20%) did know of any method of cervical screening. In terms of the frequency for cervical cancer screening, (174, 66%) did not know, only 48, 18%) reported that it should be once every three years, followed by (24, 9%) who indicated it should take place once a year, and lastly (19, 7%) who indicated it should be once every five years. Screening services should be made available, and follow-up is important and should be encouraged after a negative test.

### 4.6. Knowledge Regarding the Eligibility for Cervical Cancer Screening

The findings revealed that (112, 42%) respondents were not knowledgeable about who is eligible for cervical cancer screening and (79, 30%) indicated that women of 30 years and above are eligible for cervical cancer screening, with a small percentage citing elderly woman (35, 13%) and women aged less than 21 years (25, 9%). Lastly, some mentioned others (17, 6%). Eligibility should also be related with risk factors and the current epidemiological reports of what the current status quo of the country may be. (Table 2).

**Table 2.** Perceptions of female nurses towards cervical cancer screening.

| Questions | SA | | A | | N | | D | | SD | |
|---|---|---|---|---|---|---|---|---|---|---|
| | N | % | N | % | N | % | N | % | N | % |
| Do you believe cervical cancer is a killer in the country? | 112 | 42.4 | 12 | 4.5 | 115 | 43.6 | 14 | 5.3 | 11 | 4.2 |
| Do you believe screening helps the prevention of carcinoma of the cervix? | 147 | 55.7 | 32 | 12.1 | 63 | 23.9 | 7 | 2.7 | 15 | 5.7 |
| Do you believe screening causes no harm to the client? | 13 | 4.9 | 6 | 2.3 | 195 | 73.9 | 8 | 3.0 | 42 | 15.9 |
| If screening is paid, will you screen? | 16 | 6.1 | 64 | 24.2 | 42 | 15.9 | 68 | 25.8 | 74 | 28.0 |
| Do you perceive any adult woman including you can acquire cervical carcinoma? | 125 | 47.3 | 21 | 8.0 | 60 | 22.7 | 32 | 12.1 | 26 | 9.8 |
| Do you believe screening can detect cervical changes before it becomes cancer? | 123 | 46.6 | – | – | 2 | 0.8 | 61 | 23.1 | 78 | 29.5 |
| Do you think going through the screening procedure is an embarrassment? | 188 | 71.2 | 56 | 21.2 | 7 | 2.7 | 6 | 2.3 | 7 | 2.7 |
| If you were screened, would you allow male doctors to examine your cervix? | 62 | 23.5 | 41 | 15.5 | 17 | 6.4 | 16 | 6.1 | 128 | 48.5 |
| If you develop cervical cancer, will you consult doctors without being scared? | 26 | 9.8 | 45 | 17.0 | 63 | 23.9 | 48 | 18.2 | 82 | 31.1 |

Table 2 indicate a large number of female nurses (112, 42.4%) strongly agreed that cervical cancer is killer, while a considerable number (115, 43.6%) remained neutral/undecided, with lowest percent strongly disagreeing (25, 8.7%). However, (179, 67.8%) indicated that screening is key in preventing carcinoma of the cervix, whilst (63, 23.9%) of them remained undecided and (22, 8.4%) did not agree. The results shows that respondents are aware that cervical cancer contributes to mortalities and understand the benefits of screening. An alarming number of respondents disagreed and strongly disagreed (142, 53.8%) to be

screened if it was paid for, and (118, 44.6%) perceived themselves as not vulnerable to acquiring cervical carcinoma. Furthermore, the majority (244, 92.4%) of respondents indicated that screening for cervical cancer was embarrassing, and (193, 73.2%), revealed that they would be scared to consult about screening if they develop cervical cancer. This is an indication of ignorance in that they are embarrassed and fear the unknown. One-hundred and three (39%) of the respondents agreed to be screened by male doctors; however, (128, 48.5%) strongly disagreed and (17, 6.4%) were undecided. Respondents as health care professionals and part of the multidisciplinary team displayed a negative attitude towards screening practices by citing embarrassment and gender preference influence their health and well-being. even as health professionals and part of a multi-disciplinary tea Attitudes are great contributor to positive health perceptions, while a negative perception can contribute to a lack of self-care/poor perception of health.

### 4.7. Participation in Cervical Cancer Screening

According to the study findings, (218, 83%) had been screened for cervical cancer and only (46, 17%) had not been screened at all. Out of the 46 who had not been screened, most (*n* = 30) were enrolled nursing auxiliaries, eleven (*n* = 11) were enrolled nurses, and five (*n* = 5) were registered nurses, which implies that the lower categories of nurses are either not informed or ignorant. The majority (190, 80%) of them were last screened more than three years prior to the study, with the some (27, 10.2%) being screened within the last three years. This is of concern to health professionals. Health behaviour should be a lifestyle, and reproductive health especially should be given a personal priority.

### 4.8. Barriers to Screening for Cervical Cancer

For the minority (46, 17%) who had not been screened for cervical cancer, the results show the reason for not screening. The most cited reason was that they think they are healthy (14, 31%), it is embarrassing (13, 28%) to be screened, and, lastly, due to fear of positive results (14, 31%). However, a minority (4, 2%) cited the unavailability of services, fear of pain (22, 10%), and carelessness (4, 2%).

## 5. Discussion of Findings

The study revealed a better number of professional nurses had diplomas, although a few had degrees, and this level of qualification is therefore expected to enhance knowledge regarding cervical cancer and screening. Educational programs differ according to the competency frameworks stipulated in the programs and the level of qualification standard, which indicates the minimum credits and type of content to be provided at that level. Utoo et al. [15] consistently found that women with low levels of education did not see the need for cervical cancer screening, unlike those who were educated and knew the consequences of not screening. The variance in information regarding susceptibility to cervical cancers among these professional groups is related to the type of training provided during college education and, probably, the lack of ongoing mentorship after graduation [16]. Continuing training and capacity development is required and should be structured and formalised, as there has been an explosion of information in the media. The internet has been a great source of information, as the respondents indicated, but such knowledge needs to be scrutinized for relevancy. Korp [17] also suggest that Internet-related information is good; consumers should review it for its relevance to health promotion.

Although the study respondents showed a better understanding on some knowledge aspects, it is worrying that the study revealed low levels of knowledge on prevention and eligibility for cervical cancer screening, as 66% did not know the frequency to be screened, with 18% indicating once in three years, (24, 9%) yearly, and (19, 7%) once in 5 years. Forty-two percent of female nurses indicated they do not know who is eligible, (79, 30%) indicated that it was women above 30 years, and (35, 13%) indicated that it was those who were 21 yrs.

Since the respondents were not aware of the eligibility criteria and frequencies, it suggest that they are not genuinely participating in cervical screening services. As evidenced by the other literature, including [18], nurses' awareness and insight did not translate to a change in attitude and proper utilization of screening services. The present study findings indicate that a lack of knowledge can dissuade nurses from taking cervical cancer screening. Furthermore, most of the study respondents had five children and above, which places them at risk, as a parity is a risk factor. Respondents were able to indicate the risk factors of cervical cancer, such as early sexual intercourse, multiple sexual relationships, and acquiring HPV. Prevention is better than cure, and understanding the risks can better shape one's sexual practices, such as the use of condoms for safe sex and regular pap smears. Ramathuba et al. [19] indicated that most of these women are in a triad of sexual relationships as most Venda men have more than one wife; culturally, polygamy is still being practised and those not practising it officially, are involved in extramarital relationships. Multiparity is associated with cervical abnormalities resulting from cervical injury during delivery, which places women at risk of cancer of the cervix [20]. The present study findings indicate that respondents were all employed female nurses, thus indicating the affordability of cervical screening services to them; however, they indicated not to be keen on the screening if it came with the at a little fee. Lyimo et al. [21] cemented this assertion by propounding that unemployed woman relied on their partners to finance reproductive health services. Unemployment was found to be a deterrent in the affordability of cervical cancer screening among single women, and employed women could access reproductive services due to affordability.

However, Akinyemiju [22] highlighted no significant relationship between screening and health insurance, even though a higher percentage of financially able women were screened than their counterparts. The findings concur with the authors, as lower categories of nurses did not participate in the screening at any given time.

The results indicate that respondents' attitudes to some cervical cancer screening practices was favourable as they revealed that cervical cancer is a killer, that screening is key, and that they feel vulnerable to it. These findings were also previously observed in a study conducted in Ethiopia by Gebreegziabher et al. [10], where many participants acknowledged that screening was essential and necessary to be considered in women's health. These results are similar to the Republic of Korea and south-eastern Nigeria Tekalegn et al. [20], where lack of responsiveness contributed to most nurses not deciding on cervical cancer screening. The authors in [23,24] further posited that the lack of awareness campaigns and poor dissemination of information about the disease have a negative effect in the community. As evidenced by [18], the awareness that cervical cancer is dangerous did not change the nurses' attitude towards adherence to the screening services.

This present study has demonstrated that female nurses' negative attitudes could deter them from taking up cervical cancer screening.

Health care professionals are at the forefront of ensuring that women in communities are encouraged and sensitized to come for screening; however, if health professionals themselves are not practicing what they preach, it means the mandate of the department as in the National Health Plan will not be realized nor will the sustainable development goals be achieved in ensuring women's health. Cultural beliefs and norms negatively impact on the uptake of screening services, these cultural issues should be addressed as the health care professionals are women of the same cultural group. This is supported by the authors in [25] who reported that a significant decrease was observed in mammography and cervical cancer screening associated with male internists and family physicians but not with male gynaecologists in the United States, and a significant physician sex effect of $p < 0.05$ on screening from family practices. Furthermore, Osei et al. [26] reported gender preferences as a matter of concern in the invasion of personal privacy, while other findings showed that affluent communities were providing self-sampling kits to women who did not attend regular cervical cancer screening to increase their participation.

The chi-square test was carried out to determine the relationship between knowledge about cervical cancer and the pap smear test. The results ($p < 0.05$) indicated that knowledge about cervical cancer has a significant effect on pap smear uptake, suggesting that those who were knowledgeable were likely to undertake a pap test, compared to nurses who were not knowledgeable. In summation, the authors in [27] revealed that increased knowledge about cervical cancer and the importance of screening leads to a positive perception regarding health promotion and disease prevention and creates an accurate base for correct beliefs about cervical cancer tests.

## 6. Limitation of the Study

The study has some limitations which include resource constraints (time, money for data collection), and that is why the study was conducted in one district in four hospitals, so the results cannot be generalized.

Limitations due to self-rated reports can distort the study results as respondents may rate their opinion instead of personal reality. The cross-sectional study's intrusive nature may not establish temporal relationships between exposure and outcome measures.

## 7. Recommendation

Policy measures should target the awareness and participation of all categories of female nurses concerning cervical cancer screening. New information about cervical cancer screening should be updated and infused into the curricula for nurses. Practical cervical cancer refresher courses for capacity development sessions should be arranged continuously. The study should be replicated in other settings to evaluate knowledge levels that may assist in determining the national indicator of awareness. Modifications in the facilitation of cervical cancer and the involvement of doctors should be advocated for.

## 8. Implications for the Study

The findings will be useful to district managers and the Ministry of Health in the province to plan strategies involving healthcare professionals in matters of strategic importance for the health. Attrition that occurs as a result of mortalities and morbidities from cancers can be prevented as they can impact the department negatively.

## 9. Conclusions

The present study therefore can conclude that the knowledge of nurses regarding cervical cancer is fair; however, their attitudes is not favourable to cervical cancer screening. Nurses should be proactive in cervical cancer screening to influence women in communities to participate.

**Author Contributions:** L.M. (University of Venda) is the author of this article, which is based on her master's dissertation; D.U.R. (University of Venda) was the supervisor of the thesis work and manuscript draft, and M.S.M. (University of Venda) was co-supervisor. L.M, D.U.R. and M.S.M. drafted and edited the final manuscript. All authors have read and agreed to the published version of the manuscript.

**Funding:** This research received no funding.

**Institutional Review Board Statement:** The project has been approved by Research Ethics Committee of the University of Venda. The approval number is (SHS/21/PDC/07/0505).

**Informed Consent Statement:** Informed consent was obtained from all subjects involved in the study.

**Data Availability Statement:** All data supporting this manuscript have been made available. All data generated or analysed during this study are included in this article.

**Acknowledgments:** The authors are grateful to the nursing staff who participated in this research study.

**Conflicts of Interest:** The authors declare that they have no financial or personal relationship which may have inappropriately influenced them in writing this article.

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
