# Peer review of "Factors Affecting Participation in Cervical Screening by Female Nurses in Public Health Institutions in Vhembe District, Limpopo Province"

_nursrep, doi:10.3390/nursrep13010039_

Round 1

Reviewer 1 Report

Dear authors, thank you so much for presenting this excelent manuscpript, resulting for a high importance issue studied by your team. 

I have a few suggestions to improve even more your manuscript: 

1) In the abstract, line 15, consider put a quote about the importance of cervical screening to introduce the next sentence.

2) It would be good to have an appendix with the "2.6. Measurement instrument in data collection" described between lines 160-173.

3) Consider formatting the letter size on tables. It's too big!z

4) In the sentence on line 324 "The present study findings indicate that all respondents were all employed female 324 nurses, thus indicating the ability to afford the cost of reproductive health care services 325 including cervical screening services, however they indicated not to be keen in the screen- 326 ing if it came with the at a little fee." Consider to erase one of the "all".

A few litle more steps and the manuscrips becomes perfect. 

Good work! 

Author Response

Thank you for valuable comments

Reviewer 2 Report

Dear Authors

Thank you for the opportunity to read your manuscript, which I read with great interest.

The manuscript is well structured, however, it needs some changes that will improve it significantly. Below you will find some points in the manuscript that need clarification, refinement, reanalysis, rewriting and/or additional information and suggestions on what can be done to improve it.

Title - a bit long, should be up to 17 words

Abstract - descriptors should be reviewed and match DeCS/MeSh

Section 1 (Introduction) - this section needs some adjustments, as some information and/or points are missing or unclear, and should be included or better written, I will present some items:

- What is the importance of doing this research/contribution it brings to the literature in the field?

- Why should readers be interested?

- What problem/ gap does this research solve/fill?

- How will the proposed study remedy this deficiency/lacuna/problem and provide a unique contribution to the literature.

- It seems to me that the syllabus of nursing courses, referred to as professional level, could also be addressed

- Objective of the study, which should be in line with that presented in the abstract.

Section 2 (Materials and Methods) - in this section some points should be clarified and improved and included, namely:

- The type of study, will it not be a case study?

- Inclusion and exclusion criteria to be included in the study

- The instrument of data collection should be better explained

- The statistical analysis protocol should be presented

Section 3 (Results) - Presentation of results seems reasonable to me.

- Tables and figures should follow the guidelines of the journal

Section 4 (Discussion) - in the discussion a bridge could also be made between the level of training (Study Plan) and the nurses' knowledge.

There is talk of the chi square test, but in the data analysis it is not clear that this test was done.

Section 5 (conclusion) - Strategies could be presented to reduce the problem under study, namely by reviewing the syllabus of nursing courses, where, by improving this aspect, nurses' knowledge is improved, thus helping to inform the population and even motivate them about the importance of screening.

Author Response

Thank you for the inputs
